# Oral Bacteria and Their Antibiotic Susceptibilities in Taiwanese Venomous Snakes

**DOI:** 10.3390/microorganisms10050951

**Published:** 2022-04-30

**Authors:** Po-Chun Chuang, Wen-Hao Lin, Yi-Chun Chen, Chun-Chih Chien, I-Min Chiu, Tein-Shun Tsai

**Affiliations:** 1Department of Emergency Medicine, Kaohsiung Chang Gung Memorial Hospital, Kaohsiung 833401, Taiwan; zhungboqun@gmail.com (P.-C.C.); ray1985@cgmh.org.tw (I.-M.C.); 2College of Medicine, Chang Gung University, Taoyuan 333323, Taiwan; 3Institute of Wildlife Conservation, National Pingtung University of Science and Technology, Pingtung 912301, Taiwan; chrislin840518@gmail.com; 4Division of Infectious Diseases, Department of Internal Medicine, Kaohsiung Chang Gung Memorial Hospital, Kaohsiung 833401, Taiwan; sonice83@yahoo.com.tw; 5Department of Laboratory Medicine, Kaohsiung Chang Gung Memorial Hospital, Kaohsiung 833401, Taiwan; jessica0307@cgmh.org.tw; 6Department of Biological Science and Technology, National Pingtung University of Science and Technology, Pingtung 912301, Taiwan

**Keywords:** pit vipers, elapids, snakebites, wound infection, antibiotic susceptibility, neglected diseases

## Abstract

Wound infections after venomous snakebites are clinically important. Information regarding the nature and antibiotic susceptibilities of snake oral bacterial flora could support empiric antibiotic therapy. Wild venomous snakes were collected from southern Taiwan: a total of 30 each of *Bungarus multicinctus*, *Naja atra*, *Protobothrops mucrosquamatus,* and *Trimeresurus stejnegeri*; 3 *Deinagkistrodon acutus*; and 4 *Daboia siamensis*. The species and antibiotic susceptibilities of their oral bacteria were determined. Aerobic gram-negative bacteria, especially *Pseudomonas aeruginosa* and *Proteus vulgaris*, were the most abundant. *Proteus vulgaris* were more abundant in *B. multicinctus*, *N. atra,* and *P. mucrosquamatus* than in *T. stejnegeri* (40%, 43.3%, and 40% vs. 13.3%, respectively). The gram-negative species were less susceptible to first- and second-generation cephalosporins and ampicillin-sulbactam than to third-generation cephalosporins, fluoroquinolones, carbapenems, or piperacillin-tazobactam. The most abundant aerobic gram-positive species cultured was *Enterococcus faecalis*, which was more abundant in *N. atra* than in other snakes (*p* < 0.001) and was highly susceptible to ampicillin, high-level gentamicin, penicillin, teicoplanin, and vancomycin. *Bacteroides fragilis* and *Clostridium* species were the most common anaerobic bacteria. The anaerobic organisms were highly susceptible to metronidazole and piperacillin. As a reference for empiric antimicrobial therapy, third-generation cephalosporins, fluoroquinolones, carbapenems, or piperacillin-tazobactam can be initiated in venomous snakebites wound infections.

## 1. Introduction

In tropical and subtropical countries, snakebites are a serious issue and a priority neglected disease [1,2,3,4]. Worldwide, there are 1.2–5.5 million snakebites each year, of which 125,000 result in death or disability, especially in Southeast Asia [5,6]. There are six major venomous snakes in Taiwan, specifically *Naja atra*, *Bungarus multicinctus*, *Protobothrops mucrosquamatus*, *Trimeresurus stejnegeri*, *Deinagkistrodon acutus,* and *Daboia siamensis* [7,8,9]. About 800–1000 people experience venomous snakebites in Taiwan every year [10]. Antivenom administration is the standard treatment for snake bites. However, venomous snakebites can cause tissue necrosis and infection by snake oral flora [11]. Wound infection can even cause sepsis, necrotizing fasciitis, or both [11,12]. Traditional snakebite research has focused on the administration of antivenom; related microbial infections have scarcely been investigated [13]. These wound infection pathogens may come from the environment, surrounding skin, or oral mucous membranes of a venomous snake [14]. Mixed infections by bacteria from the snake’s mouth or patient’s skin could occur in wounds caused by bites from venomous snakes [5,15,16]. Some corresponding bacterial colonies have been observed in wound cultures, including gram-negative, gram-positive, and anaerobic bacteria [12,17]. Identifying snake oral microbes is important for understanding the causes of secondary infections [16] and assisting in an effective choice of antibiotics by providing data, which enables the most appropriate treatment of wound infection in snakebite patients [12,13,15].

Krishnankutty et al. [12] combined next-generation DNA sequencing methods (gene high-throughput analysis) with bioinformatic analysis tools to comprehensively and thoroughly explore the composition, function, and evolution of bacterial communities. Snake oral and venom floras vary with geographic area, snake species, and oral health [11,18]. Researchers have reported oral flora, including gram-negative bacteria (*Aeromonas hydrophila*, *Citrobacter diversus*, *Citrobacter freundii*, *Enterobacter* spp., *Escherichia coli*, *Klebsiella pneumoniae*, *Morganella morganii*, *Proteus* spp., *Providencia rettgeri*, *Pseudomonas aeruginosa*, *Pseudomonas pseudoalcaligenes*, *Salmonella typhi*, *Salmonella arizonae*, *Serratia* spp., *Shewanella putrefaciens*, *Shigella* spp., and *Yersinia enterocolitica*), gram-positive bacteria (*Enterococcus* spp., *Bacillus* spp., β-hemolytic streptococci, coagulase-negative staphylococci, *Micrococcus* sp., Group D *Streptococcus aureus,* and *Streptococcus viridans*) and anaerobic bacteria (*Clostridium perfringens*, *Streptococcus* spp., and *Bacteroides* spp.) [13,19]. Studies have shown that snake oral microbes are related to those excreted by their prey, perhaps being excreted when the prey is bitten. However, recent studies have found that the oral cavity flora of snakes is significantly different from the fecal flora of the prey [20]. The results of 16S ribosomal gene sequence identification are not consistent with those derived using biochemical (VITEK platform) analysis [21]. Based on previous study results [5,11,13,18,22,23], the possible secondary infecting organisms in venomous snakebites are listed in Table 1.

Mao et al. [18] found that *M. morganii* was the most common species in 112 cases of wound infections following bites from *N. atra* treated at Taichung Veterans General Hospital. It was followed by *Proteus* spp., *A. hydrophila*, *P. aeruginosa*, and *Providencia* spp. *Enterococcus* spp. were the most common gram-negative bacteria, whereas *Bacteroides* and *Shewanella* spp. were the most common anaerobic bacteria. The study also tested the susceptibilities of different strains to various antibiotics and reported that the bacteria showed resistance to first- and second-generation cephalosporins and fluoroquinolones. However, among these hospital patients, there may have been exposure to multiple antibiotics and to nosocomial pathogens during the period between getting bitten and having wound bacterial infection cultures performed [18]. In addition, the organisms colonizing other important venomous snake species in Taiwan were not analyzed. Studies have reported that bites from *P. mucrosquamatus*, *T. stejnegeri*, and *D. acutus* can cause a wound infection that may require surgical intervention [24,25].

This study aimed to analyze the species, composition, and antibiotic susceptibilities of bacterial clumps in the oral cavities of six important venomous snakes in southern Taiwan. These results can be used as a reference for clinical antimicrobial therapies.

## 2. Materials and Methods

### 2.1. Sample Collection

Wild venomous snakes, namely *B. multicinctus*, *D. acutus*, *D. siamensis*, *N. atra*, *P. mucrosquamatus*, and *T. stejnegeri*, were collected from Kaohsiung City and Pingtung County in southern Taiwan. Before sampling, the snakes did not eat or receive any drugs or antibiotics. In the snake room, the researcher carefully grasped and fixed the head and neck of each snake, opened its mouth with a sterilized mouth prop, smeared the inside of the mouth with a sterilized cotton swab, and placed the cotton swab in a sterile sample storage tube (108C and 114C, COPAN Diagnostics Inc., Murrieta, CA, USA). The samples were immediately transferred to the Department of Laboratory Medicine at the Kaohsiung Chang Gung Memorial Hospital for bacterial isolation and identification. The study was approved (approval number NPUST-109-023) by the Animal Care and Use Committee of National Pingtung University of Science and Technology.

### 2.2. Bacterial Isolation, Identification, and Antibiotics Susceptibilities

Swab specimens were plated directly onto trypticase soy agar II with 5% sheep blood agar, Levine EMB agar, and Columbia CNA agar. The cultures were incubated at 35 °C in 5% CO_2_ for 5 days. These specimens were simultaneously inoculated onto CDC ANA agar, phenylethyl alcohol agar, and LKV agar, which were immersed in thioglycolate broth. These cultures were incubated in a Concept Plus Anaerobic Chamber and then placed at 35 °C in ambient air for 5 days. The unknown bacterial colonies were identified using matrix-assisted laser desorption ionization time of flight mass-spectrometry (Microflex LT, Bruker Daltonik GmbH, Bremen, Germany) [26]. If there were more than six different bacteria species, we identified the most dominant six species and presented those species that have been reported as potentially secondary infectious in snakebite wound infections (Table 1). Antibiotic susceptibility of the isolated aerobic microbes was determined using the Kirby–Bauer diffusion method (BBL, Muller-Hinton II agars; Becton Dickinson Microbiology Systems, Cockeyville, MD, USA). The antimicrobial susceptibilities of anaerobic bacteria were determined using the agar dilution method. The susceptibility testing standards were those from the Clinical and Laboratory Standards Institute’s guidance [27,28].

### 2.3. Statistics

Data on the bacteria identified from the mouths of venomous snakes are presented as numbers with percentages. The Chi-square and Fisher’s exact tests were used to analyze the data. A two-tailed test with a *p*-value < 0.05 was considered statistically significant. All statistical analyses were performed using SPSS for Windows, version 22.0 (released 2013, IBM Corp., Armonk, NY, USA).

## 3. Results

### 3.1. Percentages and Numbers of the Isolated Bacteria

A total of 127 wild venomous snakes, specifically 30 each of *B. multicinctus*, *N. atra*, *P. mucrosquamatus*, and *T. stejnegeri*; 3 *D. acutus;* and 4 *D. siamensis* were collected. A total of 510 potentially infectious organisms were isolated, consisting of 181 anaerobic, 86 aerobic gram-positive, and 243 aerobic gram-negative bacterial species. The percentages and numbers of the isolated organisms are shown in Figure 1.

### 3.2. Comparisons of the Bacteria from Different Snake Species

Table 2 reports the organisms cultured from the oral cavities of the six venomous snake species. No interspecies differences in isolation rates were statistically significant in relation to anaerobic bacteria. In respect of aerobic gram-positive species, *T. stejnegeri* had a higher frequency of *Enterococcus casseliflavus* present in the oral cavity than *B. multicinctus*, *N. atra*, and *P. mucrosquamatus* (13.3% vs. 3.3%, 0% and 0%, respectively; *p* = 0.029); *N. atra* had the highest frequency of *Enterococcus faecalis* (100%, *p* < 0.001); *P. mucrosquamatus* carried *Staphylococcus sciuri* more frequently than *B. multicinctus*, *N. atra*, and *T. stejnegeri* (13.3% vs. 0%, 0% and 0%, respectively; *p* = 0.012). With respect to the aerobic gram-negative bacteria, the *C. freundii* complex was cultured from *T. stejnegeri* more often than from other venomous snakes (46.7%; *p* = 0.006); the *Enterobacter cloacae* complex was cultured from *P. mucrosquamatus* and *T. stejnegeri* more frequently than from *B. multicinctus* and *N. atra* (20% and 23.3% vs. 3.3% and 3.3%, respectively; *p* = 0.035); *Proteus vulgaris* was more common in the oral cavities of *B. multicinctus*, *N. atra*, and *P. mucrosquamatus* than in *T. stejnegeri* (40%, 43.3%, and 40% vs. 13.3%, respectively; *p* = 0.029). The five most common organisms in the oral cavities of *B. multicinctus*, *N. atra*, *P. mucrosquamatus*, and *T. stejnegeri* are shown in Figure 2.

### 3.3. Antibiotic Susceptibility of Isolated Bacteria

The susceptibilities of the bacteria isolated from the oral cavities of venomous snakes are presented in Table 3. The anaerobic organisms showed 100% susceptibility to metronidazole. *Bacteroides* species showed lower susceptibilities to clindamycin, penicillin, and ampicillin-sulbactam. *Clostridium* species showed high susceptibility to antibiotics. Among aerobic gram-positive organisms, *E. faecalis* was the most common organism and showed 100% susceptibilities to ampicillin, high-level gentamicin (120 mcg/mL), penicillin, teicoplanin, and vancomycin. *E. casseliflavus* was resistant to vancomycin. *Staphylococcus sciuri* was resistant to clindamycin, erythromycin, oxacillin, and penicillin. Among aerobic gram-negative organisms, *P. aeruginosa* was the most common and was highly susceptible to amikacin, ciprofloxacin, ceftazidime, cefepime, cefoperazone-sulbactam, colistin, gentamicin, imipenem, levofloxacin, meropenem, and piperacillin-tazobactam. *Proteus vulgaris* had the second highest culture rate and was less susceptible to cefazolin and cefuroxime (2.3% and 7%, respectively). *Acinetobacter* spp., *A. hydrophila*, *C. freundii* complex, *E. cloacae* complex, and *K. pneumoniae* were also frequently cultured. *K. pneumoniae* had lower susceptibilities to ciprofloxacin and cefazolin (78.6% and 50%, respectively).

## 4. Discussion

In this study, gram-negative bacteria dominated the oral cavities of venomous snakes, followed by anaerobic organisms and gram-positive bacteria (48%, 35% and 17%, respectively; Figure 1). This differs from the predominance of gram-positive bacteria in wound infections associated with snakebites [11]. By isolating directly from the oral cavities of snakes, rather than from bite wounds, we avoided contamination by epidermal or nosocomial bacteria. Gram-negative organisms have previously been shown to be more prevalent in venomous snakes’ oral cavities [13]. The community composition and bacterial susceptibilities of the oral cavity flora will vary based on the snake’s living environment and the fecal flora of the prey it consumes [29,30]. However, Zancolli et al. [31] have claimed that the prey’s fecal bacterial flora has little bearing on the oral flora of snakes.

Thirty species of microbes cultured from the oral cavities of six venomous snakes were identified (Table 2). The two most common anaerobic organisms were *Bacteroides fragilis* and *Clostridium* spp., and the isolation frequencies of these did not differ significantly between the six species of venomous snakes. Among the aerobic gram-positive species, *E. faecalis* had a higher frequency in the oral cavities of *N. atra* than in those of other snakes (*p* < 0.001). Among the aerobic gram-negative organisms, *P. aeruginosa* and *P. vulgaris* were the two species isolated most frequently, and *P. vulgaris* had a higher rate of isolation from *B. multicinctus*, *N. atra*, and *P. mucrosquamatus* than from *T. stejnegeri* (40%, 43.3%, and 40% vs. 13.3%, respectively; *p* = 0.029).

The rate of secondary wound infection varies among patients bitten by different venomous snakes. In Taiwan, *N. atra* has the highest infection rate at approximately 80.9% [7]. Mao et al. [32] identified three important organisms based on comparisons between oral bacteria from *N. atra* and wound bacterial cultures from envenomed patients, namely *M. morganii*, *P. vulgaris*, and *Proteus mirabilis.* In the current study, these three bacteria were isolated from the oral cavities of venomous snakes, mainly from *N. atra* and *P. mucrosquamatus*. However, the most commonly isolated organisms from the oral cavities of *N. atra* were *E. faecalis, Clostridium* spp., *P. aeruginosa*, *B. fragilis*, and *P. vulgaris* (Figure 2b), which should also be considered as pathogens.

Recent studies have reported that the secondary wound infection rates in patients envenomed by *P. mucrosquamatus* and *T. stejnegeri* are approximately 25% and 12%, respectively [9,33,34]. When envenomed by *B. multicinctus*, the infection rates are much lower. Among the less common venomous snakebites, wounds caused by *D. acutus* are more likely to become infected and require antibiotics or even surgery than wounds caused by *D. siamensis* [25,35]. There is not enough information to determine wound infection rates for other snakebites, which are less frequently reported in Taiwan, such as those caused by *Trimeresurus gracilis* and *Ovophis makazayazaya* [36,37]. The nature and susceptibilities of the oral bacterial flora of these two venomous pit vipers need to be determined in future studies.

There were fewer drug-resistant strains in this study (Table 3). This may be because we swabbed the oral cavities of wild snakes and immediately sent the samples for bacterial isolation and identification. Panda et al. [16] documented diverse bacteria predominantly in the oral cavity of *Daboia russelii* from both captive and wild sources. They took samplings not immediately after capture (i.e., 7 days later) and reported that most of the isolates were resistant to antibiotics, including penicillin, amoxicillin-clavulanic acid, and oxacillin while sensitive to imipenem, amikacin, ciprofloxacin, gentamicin, and azithromycin. The antibiotic susceptibilities of the bacteria from their study should have both natural and artificial causes. As in another previous study that was conducted in northern and central Taiwan and that analyzed the bacteriology of *N. atra* bite wounds [18], we found a lower susceptibility rate for gram-negative organisms to cefazolin, cefuroxime, and ampicillin-sulbactam. Mao et al. [18] observed the antibiotic resistance of *Pseudomonas* spp. to fluoroquinolone and of *Enterococcus* spp. to penicillin, which was not observed in our study. The difference could be due to interference from secondary infection and previous antibiotic exposure.

Clinically, some physicians in Taiwan use first-generation cephalosporins or dicloxacillin as prophylactic antibiotics [24]. However, the susceptibilities of venomous snake oral cavity bacteria to cefazolin and oxacillin were too low to make this an effective approach. It is more reasonable to administer the appropriate antibiotics to patients with signs of infection than prophylactic antibiotics to envenomed patients. *P. aeruginosa* should be considered because it is the most common gram-negative organism in the oral cavities of venomous snakes. Physicians can administer third-generation cephalosporins, fluoroquinolones, carbapenems, or piperacillin-tazobactam to patients with evidence of a wound infection.

In terms of limitations, the study only reports the possible infecting organisms, perhaps overlooking other organisms not noted by previous studies, and next-generation sequencing analysis could be conducted to comprehensively reveal bacterial diversity. Secondly, although isolating the organisms directly from the oral cavities of venomous snakes reduces the influence of skin flora and nosocomial infections, physicians still need to consider secondary skin and nosocomial infections after snake envenomation if patients are hospitalized. Thirdly, the number of samples was small, especially for the rarer venomous species *D. acutus* and *D. siamensis*.

## 5. Conclusions

Patients who are bitten by venomous snakes may develop wound infections caused by bacteria from the snake’s mouth or the patient’s skin. This study determined the pattern of the oral bacterial flora and their sensitivity to antibiotics in freshly captured native snakes in Taiwan. Among the oral bacterial flora of venomous snakes, aerobic gram-negative organisms are the most frequently isolated, among which *P. aeruginosa* and *P. vulgaris* are the most common. In addition, the *C. freundii* complex, *Acinetobacter* spp., and *A. hydrophila* were noteworthy. Gram-negative organisms showed low sensitivities to first- and second-generation cephalosporins and ampicillin-sulbactam. The most common aerobic gram-positive organism was *E. faecalis*, which had a higher isolation rate in the oral cavities of *N. atra* than in those of other snakes and was highly susceptible to ampicillin, high-level gentamicin, penicillin, teicoplanin, and vancomycin. Most of the isolated anaerobic bacteria had relatively high susceptibilities to metronidazole and piperacillin. As a reference for empiric antimicrobial therapy, physicians can initiate treating patients with snakebite wound infections before obtaining the results of wound, blood, or specimen culture. Since the most abundant oral flora of wild venomous snakes are highly susceptible to third-generation cephalosporins, fluoroquinolones, carbapenems, or piperacillin-tazobactam, these antibiotics can be a beginning choice. The results of this study can be also applied in the veterinary medicine of wild snakes.

## Figures and Tables

**Figure 1 microorganisms-10-00951-f001:**
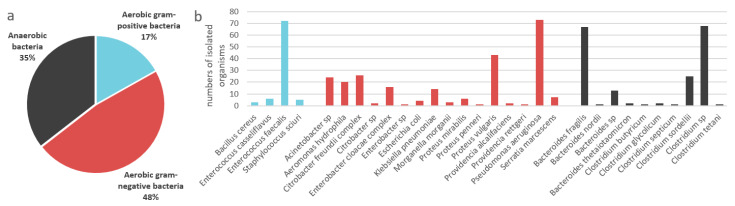
The percentages (**a**) and numbers (**b**) of isolated potentially infectious bacteria found in the oral cavities of venomous snakes.

**Figure 2 microorganisms-10-00951-f002:**
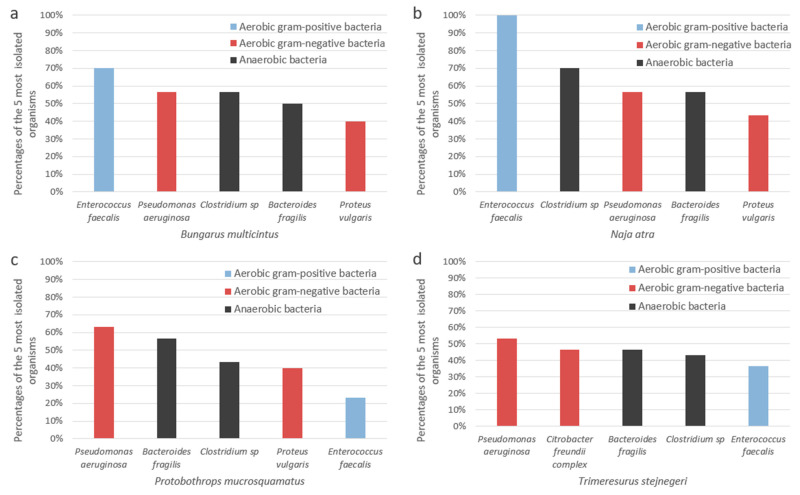
The percentages of the five most common organisms in the oral cavities of (**a**) *Bungarus multicinctus*, (**b**) *Naja atra*, (**c**) *Protobothrops mucrosquamatus*, and (**d**) *Trimeresurus stejnegeri*.

**Table 1 microorganisms-10-00951-t001:** Bacteria associated with secondary wound infection after snakebite and reported in the literature [5,11,13,18,22,23].

Aerobic Bacteria		Anaerobic Bacteria
Gram-Positive	Gram-Negative	
*Bacillus* spp.	*Acinetobacter* spp.	*Bacteroides fragilis*
*Enterococcus* spp.	*Aeromonas hydrophila*	*Bacteroides* spp.
*Staphylococcus* spp.	*Citrobacter* spp.	*Clostridium* spp.
*Streptococcus* spp.	*Enterobacter* spp.	*Peptostreptococcus* spp.
	*Escherichia coli*	
	*Klebsiella pneumoniae*	
	*Morganella morganii*	
	*Proteus* spp.	
	*Providencia* spp.	
	*Pseudomonas aeruginosa*	
	*Salmonella arizonae*	
	*Serratia liquefaciens*	
	*Serratia* spp.	
	*Shewanella putrefaciens*	
	*Yersinia enterocolitica*	

**Table 2 microorganisms-10-00951-t002:** Bacteria identified from the oral cavities of six venomous snake species.

	BM (*n* = 30)	DA (*n* = 3)	DS (*n* = 4)	NA (*n* = 30)	PM (*n* = 30)	TS (*n* = 30)	*p*-Value
**Anaerobic organisms**							
*Bacteroides fragilis*, *n* = 67	15 (50%)	2 (66.7%)	2 (50%)	17 (56.7%)	17 (56.7%)	14 (46.7%)	0.963
*Bacteroides nordii*, *n* = 1	0 (0%)	0 (0%)	0 (0%)	1 (3.3%)	0 (0%)	0 (0%)	1.000
*Bacteroides* spp., *n* = 13	3 (10%)	0 (0%)	0 (0%)	7 (23.3%)	2 (6.7%)	1 (3.3%)	0.228
*Bacteroides thetaiotaomicron*, *n* = 2	0 (0%)	0 (0%)	1 (25%)	0 (0%)	1 (3.3%)	0 (0%)	0.108
*Clostridium butyricum*, *n* = 1	0 (0%)	0 (0%)	0 (0%)	0 (0%)	1 (3.3%)	0 (0%)	1.000
*Clostridium glycolicum*, *n* = 2	1 (3.3%)	0 (0%)	0 (0%)	0 (0%)	1 (3.3%)	0 (0%)	1.000
*Clostridium septicum*, *n* = 1	0 (0%)	0 (0%)	0 (0%)	0 (0%)	1 (3.3%)	0 (0%)	1.000
*Clostridium sordellii*, *n* = 25	7 (23.3%)	0 (0%)	1 (25%)	5 (16.7%)	7 (23.3%)	5 (16.7%)	0.935
*Clostridium* spp., *n* = 68	17 (56.7%)	1 (33.3%)	3 (75%)	21 (70%)	13 (43.3%)	13 (43.3%)	0.195
*Clostridium tetani*, *n* = 1	0 (0%)	0 (0%)	0 (0%)	0 (0%)	1 (3.3%)	0 (0%)	1.000
**Aerobic gram-positive organisms**							
*Bacillus cereus,**n* = 3	0 (0%)	0 (0%)	1 (25%)	0 (0%)	1 (3.3%)	1 (3.3%)	0.158
*Enterococcus casseliflavus,**n* = 6	1 (3.3%)	1 (33.3%)	0 (0%)	0 (0%)	0 (0%)	4 (13.3%)	0.029
*Enterococcus faecalis,**n* = 72	21 (70%)	1 (33.3%)	2 (50%)	30 (100%)	7 (23.3%)	11 (36.7%)	<0.001
*Staphylococcus sciuri,**n* = 3	0 (0%)	0 (0%)	1 (25%)	0 (0%)	4 (13.3%)	0 (0%)	0.012
**Aerobic gram-negative organisms**							
*Acinetobacter* sp., *n* = 24	5 (16.7%)	0 (0%)	0 (0%)	7 (23.3%)	5 (16.7%)	5 (16.7%)	0.972
*Aeromonas hydrophila,**n* = 20	8 (26.7%)	1 (33.3%)	1 (25%)	1 (3.3%)	3 (10%)	6 (20%)	0.067
*Citrobacter freundii* complex, *n* = 26	3 (10%)	0 (0%)	0 (0%)	6 (20%)	3 (10%)	14 (46.7%)	0.006
*Citrobacter* spp., *n* = 2	0 (0%)	0 (0%)	0 (0%)	0 (0%)	1 (3.3%)	1 (3.3%)	1.000
*Enterobacter cloacae* complex, *n* = 16	1 (3.3%)	1 (33.3%)	0 (0%)	1 (3.3%)	6 (20%)	7 (23.3%)	0.035
*Enterobacter* spp., *n* = 1	0 (0%)	0 (0%)	0 (0%)	0 (0%)	0 (0%)	1 (3.3%)	1.000
*Escherichia coli*, *n* = 4	1 (3.3%)	0 (0%)	0 (0%)	2 (6.7%)	0 (0%)	1 (3.3%)	0.922
*Klebsiella pneumoniae*, *n* = 14	6 (20%)	1 (33.3%)	0 (0%)	1 (3.3%)	5 (16.7%)	1 (3.3%)	0.086
*Morganella morganii*, *n* = 3	0 (0%)	0 (0%)	1 (25%)	1 (3.3%)	1 (3.3%)	0 (0%)	0.158
*Proteus mirabilis*, *n* = 6	1 (3.3%)	0 (0%)	1 (25%)	2 (6.7%)	2 (6.7%)	0 (0%)	0.287
*Proteus penneri*, *n* = 1	0 (0%)	0 (0%)	0 (0%)	1 (3.3%)	0 (0%)	0 (0%)	1.000
*Proteus vulgaris*, *n* = 43	12 (40%)	2 (66.7%)	0 (0%)	13 (43.3%)	12 (40%)	4 (13.3%)	0.029
*Providencia alcalifaciens*, *n* = 2	1 (3.3%)	0 (0%)	1 (25%)	0 (0%)	0 (0%)	0 (0%)	0.108
*Providencia rettgeri*, *n* = 1	0 (0%)	0 (0%)	0 (0%)	0 (0%)	1 (3.3%)	0 (0%)	1.000
*Pseudomonas aeruginosa*, *n* = 73	17 (56.7%)	2 (66.7%)	2 (50%)	17 (56.7%)	19 (63.3%)	16 (53.3%)	0.999
*Serratia marcescens*, *n* = 7	1 (3.3%)	0 (0%)	0 (0%)	4 (13.3%)	2 (6.7%)	0 (0%)	0.374

Abbreviations: BM, *Bungarus multicinctus*; DA, *Deinagkistrodon acutus*; DS, *Daboia siamensis*; NA, *Naja atra*; PM, *Protobothrops mucrosquamatus*; TS, *Trimeresurus stejnegeri*.

**Table 3 microorganisms-10-00951-t003:** Antibiotic susceptibilities of bacteria isolated from oral cavities of snakes.

Antibiotics (MIC, mcg/mL)	CLI (2)		MTZ (8)		PEN (10 units)		PIP (100)	SAM (10/10)	
**Anaerobic organisms**	*n* (% of indicated susceptible)								
*Bacteroides fragilis*, *n* = 67	63 (94%)		67 (100%)		1 (1.5%)		62 (92.5%)	24 (35.8%)	
*Bacteroides nordii*, *n* = 1	0 (0%)		1 (100%)		0 (0%)		1 (100%)	1 (100%)	
*Bacteroides* sp., *n* = 13	12 (92.3%)		13 (100%)		0 (0%)		13 (100%)	12 (92.3%)	
*Bacteroides thetaiotaomicron*, *n* = 2	1 (50%)		2 (100%)		0 (0%)		2 (100%)	2 (100%)	
*Clostridium butyricum*, *n* = 1	1 (100%)		1 (100%)		1 (100%)		1 (100%)	1 (100%)	
*Clostridium glycolicum*, *n* = 2	2 (100%)		2 (100%)		2 (100%)		2 (100%)	2 (100%)	
*Clostridium septicum*, *n* = 1	1 (100%)		1 (100%)		1 (100%)		1 (100%)	1 (100%)	
*Clostridium sordellii*, *n* = 25	24 (96%)		25 (100%)		25 (100%)		25 (100%)	25 (100%)	
*Clostridium* sp., *n* = 68	68 (100%)		68 (100%)		66 (97.1%)		68 (100%)	67 (98.5%)	
*Clostridium tetani*, *n* = 1	1 (100%)		1 (100%)		1 (100%)		1 (100%)	1 (100%)	
**Antibiotics (MIC, mcg/mL)**	**AMP (10)**	**CLI (2)**	**ERY (15)**	**GEN (120)**	**OXA (1)**	**PEN** **(10 units)**	**SXT (23.75/1.25)**	**TEC (30)**	**VAN (30)**
**Aerobic gram-positive** **organisms**	*n* (% of indicated susceptible)								
*Bacillus cereus,**n* = 3		3 (100%)	3 (100%)		2 (66.7%)	3 (100%)	3 (100%)	3 (100%)	3 (100%)
*Enterococcus casseliflavus,**n* = 6	6 (100%)			6 (100%)		6 (100%)		6 (100%)	1 (16.7%)
*Enterococcus faecalis,**n* = 72	72 (100%)			72 (100%)		72 (100%)		72 (100%)	72 (100%)
*Staphylococcus sciuri,**n* = 3		3 (60%)	3 (60%)		3 (60%)	3 (60%)	5 (100%)	5 (100%)	5 (100%)
**Antibiotics (MIC, mcg/mL)**	**AMK (30)**		**CIP (5)**	**CFZ (30)**	**CXM (30)**	**CRO (30)**	**CAZ (30)**	**FEP (30)**	**SCF (75/30)**
**Aerobic gram-negative** **organisms**	*n* (% of indicated susceptible)								
*Acinetobacter* spp., *n* = 24	23 (95.8%)		24 (100%)				24 (100%)	24 (100%)	24 (100%)
*Aeromonas hydrophila,**n* = 20	20 (100%)		18 (90%)		19 (95%)	16 (80%)	20 (100%)		
*Citrobacter freundii complex*, *n* = 26	26 (100%)		26 (100%)		2 (7.7%)	26 (100%)	26 (100%)		26 (100%)
*Citrobacter* sp., *n* = 2	2 (100%)		2 (100%)	1 (50%)	1 (50%)	2 (100%)	2 (100%)		2 (100%)
*Enterobacter cloacae complex*, *n* = 16	16 (100%)		15 (93.8%)		15 (93.8%)	15 (93.8%)	15 (93.8%)		16 (100%)
*Enterobacter* spp., *n* = 1	1 (100%)		1 (100%)		1 (100%)	1 (100%)	1 (100%)		1 (100%)
*Escherichia coli*, *n* = 4	4 (100%)		4 (100%)	1 (25%)	3 (75%)	4 (100%)	4 (100%)		4 (100%)
*Klebsiella pneumoniae*, *n* = 14	14 (100%)		11 (78.6%)	7 (50%)	13 (92.9%)	13 (92.9%)	14 (100%)		14 (100%)
*Morganella morganii*, *n* = 3	3 (100%)		3 (100%)			3 (100%)	3 (100%)		3 (100%)
*Proteus mirabilis*, *n* = 6	6 (100%)		6 (100%)	2 (33.3%)	5 (83.3%)	6 (100%)	5 (83.3%)		6 (100%)
*Proteus penneri*, *n* = 1	1 (100%)		1 (100%)		1 (100%)	1 (100%)	1 (100%)		1 (100%)
*Proteus vulgaris*, *n* = 43	43 (100%)		43 (100%)	1 (2.3%)	3 (7%)	42 (97.7%)	42 (97.7%)		43 (100%)
*Providencia alcalifaciens*, *n* = 2	2 (100%)		1 (50%)	1 (50%)	2 (100%)	2 (100%)	2 (100%)		2 (100%)
*Providencia rettgeri*, *n* = 1	1 (100%)		1 (100%)		1 (100%)	1 (100%)	1 (100%)		1 (100%)
*Pseudomonas aeruginosa*, *n* = 73	73 (100%)		73 (100%)				73 (100%)	73 (100%)	73 (100%)
*Serratia marcescens*, *n* = 7	7 (100%)		7 (100%)			6 (85.7%)	6 (85.7%)		6 (85.7%)
**Antibiotics (MIC, mcg/mL)**	**CST (10)**		**ETP (10)**	**GEN (10)**	**IPM (10)**	**LVX (10)**	**MEM (10)**	**SAM (10/10)**	**TZP (100/10)**
*Acinetobacter* sp., *n* = 24	21 (87.5%)			22 (91.7%)	23 (95.8%)		24 (100%)	24 (100%)	
*Aeromonas hydrophila,**n* = 20			4 (20%)	20 (100%)		20 (100%)			
*Citrobacter freundii complex*, *n* = 26			26 (100%)	26 (100%)		25 (96.2%)		5 (19.2%)	26 (100%)
*Citrobacter* sp., *n* = 2			2 (100%)	2 (100%)		2 (100%)		1 (50%)	2 (100%)
*Enterobacter cloacae complex*, *n* = 16			16 (100%)	16 (100%)		15 (93.8%)		5 (31.3%)	16 (100%)
*Enterobacter* sp., *n* = 1			1 (100%)	1 (100%)		1 (100%)		1 (100%)	1 (100%)
*Escherichia coli*, *n* = 4			4 (100%)	4 (100%)		4 (100%)		3 (75%)	4 (100%)
*Klebsiella pneumoniae*, *n* = 14			14 (100%)	13 (92.9%)		14 (100%)		14 (100%)	13 (92.9%)
*Morganella morganii*, *n* = 3			3 (100%)	3 (100%)		3 (100%)		1 (33.3%)	3 (100%)
*Proteus mirabilis*, *n* = 6			6 (100%)	6 (100%)		6 (100%)		5 (83.3%)	6 (100%)
*Proteus penneri*, *n* = 1			1 (100%)	1 (100%)		1 (100%)		1 (100%)	1 (100%)
*Proteus vulgaris*, *n* = 43			43 (100%)	43 (100%)		43 (100%)		43 (100%)	43 (100%)
*Providencia alcalifaciens*, *n* = 2			2 (100%)	2 (100%)		1 (50%)		2 (100%)	2 (100%)
*Providencia rettgeri*, *n* = 1			1 (100%)	1 (100%)		1 (100%)			1 (100%)
*Pseudomonas aeruginosa*, *n* = 73	73 (100%)			73 (100%)	73 (100%)	73 (100%)	71 (97.3%)		73 (100%)
*Serratia marcescens*, *n* = 7			7 (100%)	7 (100%)		7 (100%)			6 (85.7%)

Abbreviations: MIC, minimal inhibitory concentration; CLI, clindamycin; MTZ, metronidazole; PEN, penicillin; PIP, piperacillin; SAM, ampicillin-sulbactam; AMP, ampicillin; ERY, erythromycin; GEN, gentamicin; OXA, oxacillin; SXT, sulfonamides-trimethoprim; TEC, teicoplanin; VAN, vancomycin; AMK, amikacin; CIP, ciprofloxacin; CFZ, cefazolin; CXM, cefuroxime; CRO, ceftriaxone; CAZ, ceftazidime; FEP, cefepime; SCF, cefoperazone-sulbactam; CST, colistin; ETP, ertapenem; IPM, imipenem; LVX, levofloxacin; MEM, meropenem; TZP, piperacillin-tazobactam.

## Data Availability

Not applicable.

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
