# Peer review of "Oral Bacteria and Their Antibiotic Susceptibilities in Taiwanese Venomous Snakes"

_microorganisms, 2022, doi:10.3390/microorganisms10050951_

Round 1

Reviewer 1 Report

The work presents a good approach. The introduction is relevant and theory based. Sufficient information about the previous study findings is presented for readers to follow the present study rationale and procedures. Overall, the methodology and the results are clear and compelling.  However, the work provides relevant evidence for the field.

Author Response

Review Report (Reviewer 1)

The work presents a good approach. The introduction is relevant and theory based. Sufficient information about the previous study findings is presented for readers to follow the present study rationale and procedures. Overall, the methodology and the results are clear and compelling. However, the work provides relevant evidence for the field.

Author's Reply to the Review Report (Reviewer 1)

Response: Thank you very much for your comments.

Reviewer 2 Report

Chuang et al. present and ex vivo/in vitro investigation designed to identify the species of bacteria present in a variety of vipers on the island of Taiwan and then subsequently determine the susceptibility of these bacteria to a battery of antibiotics. The rationale for the investigation was that secondary infection and limb loss often occur despite antivenom therapy. The species of snake assessed included Bungarus multicintus, Naja atra, Protobothrops mucrosquamatus, and Trimeresurus stejnegeri; Deinagkistrodon acutus; and Daboia siamensis. Some of the species were represented by 3-4 snakes whereas most had 30 snakes sampled. The snakes were all from the wild and were captured, with the roof of their mouths swabbed.

The samples were processed in a standard fashion, with bacterial colonies identified using matrix-assisted laser desorption ionization-time of flight mass spectrometry. Sensitivity to antibiotics was determined with agar-diffusion.

The results are clearly depicted in the numerous tables and figures which are well-done. The predominant bacteria of each species of snake were determined by classification and species. The susceptibilities to antibiotics were also displayed.

The results differed from the wound cultures of patients exhibiting infections after snake bite. As mentioned as a limitation, the bacteria present on the skin of the snake bite victim may also play a role. Nevertheless, the work guides the possibility that a different approach to antibiotic therapy after snake bite with these species should be considered.

The conclusions made for this largely descriptive work are well supported with the data presented. I have no important comments.

Author Response

Review Report (Reviewer 2)

Chuang et al. present and ex vivo/in vitro investigation designed to identify the species of bacteria present in a variety of vipers on the island of Taiwan and then subsequently determine the susceptibility of these bacteria to a battery of antibiotics. The rationale for the investigation was that secondary infection and limb loss often occur despite antivenom therapy. The species of snake assessed included Bungarus multicintus, Naja atra, Protobothrops mucrosquamatus, and Trimeresurus stejnegeri; Deinagkistrodon acutus; and Daboia siamensis. Some of the species were represented by 3-4 snakes whereas most had 30 snakes sampled. The snakes were all from the wild and were captured, with the roof of their mouths swabbed.

The samples were processed in a standard fashion, with bacterial colonies identified using matrix-assisted laser desorption ionization-time of flight mass spectrometry. Sensitivity to antibiotics was determined with agar-diffusion.

The results are clearly depicted in the numerous tables and figures which are well-done. The predominant bacteria of each species of snake were determined by classification and species. The susceptibilities to antibiotics were also displayed.

The results differed from the wound cultures of patients exhibiting infections after snake bite. As mentioned as a limitation, the bacteria present on the skin of the snake bite victim may also play a role. Nevertheless, the work guides the possibility that a different approach to antibiotic therapy after snake bite with these species should be considered.

The conclusions made for this largely descriptive work are well supported with the data presented. I have no important comments.

Author's Reply to the Review Report (Reviewer 2)

Response: Thank you very much for your comments.

Reviewer 3 Report

Review of the article: Oral Bacteria and Their Antibiotic Susceptibilities in Taiwanese Venomous Snakes

Manuscript ID: microorganisms-1658743

In my opinion, the proposed manuscript is very interesting and well prepared. All experiments were well planned and performed, modern techniques were used e.g. for identification of bacterial species. I would recommend the manuscript for publication, however I think the authors should present some more details about research methods (detailed comments are presented below).

Detailed comments

Abstract – The abstract is well prepared. The authors have presented the idea of the study and most important results. However, I do not agree with the last sentence (which is also presented in conclusions). I do not think that the medical doctors should prescribe the mentioned antibiotics for patients with secondary wound infections of venomous snakebites. The selection of the antibiotic definitely should be based on the result of the appropriate test. Of course these antibiotics can (should) be considered as “a first choice” –but their activity must be confirmed.

Introduction – well prepared and informative, no critical comments.  

Materials and methods

I would be grateful if the authors could provide some more details to the section: 2.2 Bacterial isolation, identification, and antibiotics susceptibilities

  1. why these agar media were used, which bacteria were isolated on these media;
  2. The unknown bacterial colonies were identified – how many colonies were tested ( I assume that the authors got many colonies on each plate)
  3. why authors used disc-diffusion assay for aerobic bacteria and dilution method for anaerobic microbes?

Results

Results are interesting and really well presented. I have only one remark. The full name of bacteria should be presented only once in the whole text (when it is used first time). The authors should go through the whole text and change it according to this rule.

Discussion - well prepared, no critical comments (only the same remark as above – names of bacteria).

Conclusion – the weakest part of the manuscript. Lines 250-260 is in fact repetition of the results and I do not agree with the conclusion presented between lines 260-262.

Final decision – minor revision  

Author Response

Review Report (Reviewer 3)

Review of the article: Oral Bacteria and Their Antibiotic Susceptibilities in Taiwanese Venomous Snakes

Manuscript ID: microorganisms-1658743

In my opinion, the proposed manuscript is very interesting and well prepared. All experiments were well planned and performed, modern techniques were used e.g. for identification of bacterial species. I would recommend the manuscript for publication, however I think the authors should present some more details about research methods (detailed comments are presented below).

Author's Reply to the Review Report (Reviewer 3)

Response: Thank you very much for your comments and recommendation. Please see the following list of our response to the critique on a point by point basis. The revisions in the manuscript are shown in red color.

Detailed comments

Abstract – The abstract is well prepared. The authors have presented the idea of the study and most important results. However, I do not agree with the last sentence (which is also presented in conclusions). I do not think that the medical doctors should prescribe the mentioned antibiotics for patients with secondary wound infections of venomous snakebites. The selection of the antibiotic definitely should be based on the result of the appropriate test. Of course these antibiotics can (should) be considered as “a first choice” –but their activity must be confirmed.

Response: Thank you for the recommendation. The selection of the antibiotic definitely should be based on the result of the appropriate test. One clinical problem is that the wound culture may be contaminated by skin flora and it is too late to initiate antibiotic treatment when the blood culture presents bacteremia or sepsis. Empiric antimicrobial therapy is used when antimicrobials are given to a person before the specific pathogen causing an infection is known. (When it becomes known, treatment that is used is called directed therapy.) Empiric antibiotics could therefore be initiated once cellulitis or wound infection were impressed by clinics. This article just provides “a first choice” antibiotics treatment for empiric use during beginning. To make this point clearer, the last sentence of Abstract has been revised.

Introduction – well prepared and informative, no critical comments. 

Response: Thank you.

Materials and methods

I would be grateful if the authors could provide some more details to the section: 2.2 Bacterial isolation, identification, and antibiotics susceptibilities

Response: We have provided more details to this section, as the following responses.

  1. why these agar media were used, which bacteria were isolated on these media?

Response: Trypticase Soy Agar II with 5% Sheep Blood agar is a general purpose enriched medium often used to grow fastidious organisms such as Streptococcus pneumonia. Levine EMB agar is both a selective and differential culture medium. It selectively promotes the growth of Gram-negative bacteria (inhibits Gram-positive bacteria) and aids in the differentiation of lactose fermenter and non-lactose fermenting colonies such as Escherichia coli, Morganella morganii, Proteus spp., Providencia rettgeri and Aeromonas hydrophilia, etc. Columbia CNA agar and Phenylethyl alcohol agar are selective media used for the isolation of Gram-positive bacteria (especially staphylococci and streptococci) from clinical specimens such as Enterococcus spp., b-hemolytic streptococci, Coagulase-negative Staphylococcus, etc. CDC ANA Agar is an enriched, nonselective culture medium particularly useful for the isolation and cultivation of obligate anaerobes from clinical specimens such as Clostridium perfringens and Bacteroides spp. LKV agar is an enriched, selective, and differential medium for the isolation and partial identification of obligate anaerobic Gram-negative bacilli. LKV agar is useful for the rapid isolation of Prevotella species.

  1. The unknown bacterial colonies were identified – how many colonies were tested (I assume that the authors got many colonies on each plate)?

Response: It is true that we got many colonies on each plate. The oral flora in venomous snake mouth were plenty. If there were more than 6 bacteria species, this study identified as possible colonies as we can until the most dominant six species were identified. In addition, we presented the bacteria species that were reported as potentially secondary infectious in snakebites wound infections (Table 1 in the manuscript). To make this point clearer, the sentence has been added at lines 113-116 of the manuscript.

  1. why authors used disc-diffusion assay for aerobic bacteria and dilution method for anaerobic microbes?

Response: Disk diffusion method is used routinely for testing common, rapidly growing and certain fastidious bacterial pathogens. These bacterial pathogens include Escherichia coli, Pseudomonas aeruginosa and common facultative anaerobes. The agar-dilution susceptibility test measures the antimicrobial susceptibilities of obligate anaerobic bacteria strains, because the disk diffusion method cannot be applied in measuring the antimicrobial susceptibilities of obligate anaerobes (such as Fusobacterium nucleatum). We followed the guideline of CLSI performance standards for antimicrobial disk susceptibility tests, approved standard to perform the antimicrobial susceptibility tests and interpretation of these test results.

Results

Results are interesting and really well presented. I have only one remark. The full name of bacteria should be presented only once in the whole text (when it is used first time). The authors should go through the whole text and change it according to this rule.

Response: We have carefully checked and revised the scientific names of bacteria throughout the whole text. The full names of bacteria are presented only when they are used first time.

Discussion - well prepared, no critical comments (only the same remark as above – names of bacteria).

Response: Thank you.

Conclusion – the weakest part of the manuscript. Lines 250-260 is in fact repetition of the results and I do not agree with the conclusion presented between lines 260-262.

Response: Thank you for the comments and we have revised the Conclusion. As a reference for empiric antimicrobial therapy, physicians can initiate treating patients with snakebites wound infections before obtaining the results of wound, blood, or specimen culture. Since the most abundant oral flora of wild venomous snakes are highly susceptible to third-generation cephalosporins, fluoroquinolones, carbapenems, or piperacillin-tazobactam, these antibiotics can be a beginning choice.

Final decision – minor revision
